The bacterial composition signatures of perianal abscess and origin of infecting microbes

Han Song 1
Su Wenya 2
Fan Kefeng 3
Xu Zhen 1
Xu Hai 2
http://orcid.org/0000-0003-1215-9556 Wang Mingyu 2
Li Ling 2 lingli@sdu.edu.cn
Shen Wenlong 1 swl028073@qlyyqd.com
1 Department of Anorectal Surgery, Cheeloo College of Medicine, Qilu Hospital (Qingdao) , Qingdao , China
2 State Key Laboratory of Microbial Technology, Microbial Technology Institute, Shandong University (Qingdao) , Qingdao, Shandong , China
3 Department of Respiratory Diseases, Qingdao Chengyang District People’s Hospital , Qingdao , China
Bolshoy Alexander
Electronic publication date: 2025 Jan 23
Publication date: 2025
Volume: 13
Electronic Location ID: e18855
Received 2024 Sep 19; Accepted 2024 Dec 20
Copyright: © 2025 Han et al.
Copyright year: 2025
Copyright holder: Han et al.
License: This is an open access article distributed under the terms of the Creative Commons Attribution License, which permits unrestricted use, distribution, reproduction and adaptation in any medium and for any purpose provided that it is properly attributed. For attribution, the original author(s), title, publication source (PeerJ) and either DOI or URL of the article must be cited.
License URL: https://creativecommons.org/licenses/by/4.0/

Keywords: Perianal abscess, Microbial composition, 16S rRNA gene sequencing, Source of microbes

Funding: Qingdao Key Health Discipline Development ODZDZK-2022098 National Key Research and Development Program of China 2022YFE0199800 Key R&D Program of Shandong Province 2020CXGC011305 National Natural Science Foundation of China 82271658 This work was supported by the Foundation of Qingdao Key Health Discipline Development Fund under grant number ODZDZK-2022098, the National Key Research and Development Program of China under grant number 2022YFE0199800, the Key R&D Program of Shandong Province under grant number 2020CXGC011305, and the National Natural Science Foundation of China under grant number 82271658. The funders had no role in study design, data collection and analysis, decision to publish, or preparation of the manuscript.

==============================
Background

Perianal abscess is a common anal condition primarily caused by bacterial infections, yet the precise origins of these infecting bacteria remain unclear. Understanding the distinct microbial signatures associated with periaabscesses is crucial for provide fresh ideas for disease prevention.

Materials and Methods

Samples of anal skin, feces, and abscesses were collected from a cohort of 75 patients diagnosed with perianal abscesses. The microbial composition at each site was analyzed using 16S rRNA gene sequencing to characterize the bacterial communities present.

Results

Analysis using MaAslin2 revealed distinct bacterial compositional signatures among the anal skin, feces, and abscess samples. Alpha diversity analysis indicated significant differences in bacterial diversity between abscesses, anal skin, and feces, with abscesses showing lower diversity compared to higher richness in feces. Biomarker analysis highlighted key taxa such as Bacteroides and Escherichia-Shisgella in fecal samples, and Staphylococcus and Corynebacterium in anal skin samples. The analysis of abscess samples suggested that the bacterial composition may originate from the skin, feces, and potentially other sources.

Introduction

In the field of colorectal and anal surgery, perianal abscesses and anal fistulas are among the most common diseases. Clinically, when anal glands get blocked (commonly by fecal matter, foreign bodies, or due to local trauma) and infected, mainly by bacteria, an abscess forms. This leads to perianal swelling and pain, which is diagnosed as a perianal abscess. If the abscess ruptures spontaneously or persistent infection remains after drainage, it often progresses into an anal fistula. Significantly, these two diseases frequently coexist, posing a major clinical challenge (Vogel et al., 2016). The typical treatment for perianal abscesses is prompt incision and drainage. Alarmingly, between 30% and 70% of patients with perianal abscesses either already have or will develop anal fistulas. Even for those without initial fistula symptoms, around one-third will be diagnosed with it within months to years after abscess drainage (Malik, Nelson & Tou, 2010).

The formation of a perianal abscess starts with an infection in the anal glands within the anal canal. As the infection spreads, the body’s immune response kicks in, causing inflammation and the accumulation of pus, creating pressure and pain. Left untreated, the abscess may expand and rupture, potentially resulting in a fistula, an abnormal connection between the anal canal and the surrounding skin (Amato et al., 2015; Yin et al., 2023). Perianal abscesses is a common benign condition in colorectal surgery, with a 0.02% incidence rate in Europe and a 2% incidence rate in China (Zanotti et al., 2007). Perianal abscesses accounts for 8–25% of patients in the field of proctology (Tang, Kong & Tang, 2021). It can affect both infants and adults, and is more common in young adults aged 20–40, with males having a higher frequency (Sneider & Maykel, 2013).

According to previous studies, 90% of perianal abscesses are caused by infection of the anal glands (cryptoglandular infection) (Heitland, 2012). Some non-physiological risk factors can also cause perianal abscesses. Dietary factors play a significant role. A diet high in spicy, greasy, and processed foods can disrupt the normal flora balance in the anal area and irritate the anal mucosa, increasing the susceptibility to infection. Poor hygiene practices also contribute. Inadequate cleaning of the perianal region after defecation leaves fecal residue, which serves as a breeding ground for bacteria. Wearing tight and non-breathable undergarments creates a warm and humid environment conducive to bacterial growth. Additionally, certain underlying diseases are non-physiological risk factors, such as Crohn’s disease and HIV infection (Choi et al., 2018; Vos et al., 2022; Büyükaşik et al., 1998). Furthermore, anal abscesses have a significant recurrence incidence, with 15–50% of patients developing anal fistulas. Therefore, it is crucial to identify the non-physiological risk factors for infection, especially the microbial origin of perianal abscesses.

Previous studies have primarily focused on culturing abscesses to identify the bacteria present. For instance, Lu et al. (2019) found that Gram-negative bacteria, particularly Escherichia coli and Klebsiella pneumoniae, are the main pathogens causing perianal abscesses. In patients with type 2 diabetes, K. pneumoniae was more frequently detected in abscess samples (Liu et al., 2011). Previous studies employing microbial culture techniques on abscess samples from 81 adult patients yielded positive results in 74 cases, with E. coli, coagulase-negative Staphylococci, Enterococcus spp., and Staphylococcus aureus identified as the most prevalent bacteria (Ulug et al., 2010; Zhang, Zhang & Wang, 2016). Another study found that with an increase in abscess depth, the detection rate of aerobic bacteria also rises (Stelzmueller et al., 2010). Analysis of the classification and composition of intestinal microbiota in healthy individuals and patients with perianal abscess before and after surgery revealed significant differences (Yin et al., 2023).

However, traditional culture methods have led to limited comparative analyses and have exhibited various drawbacks. Advancements in sequencing techniques, such as high-throughput 16S rRNA gene sequencing, have enabled the detection of a large number of microbial community samples, thus providing rich data for analysis. Additionally, high-throughput sequencing allows for the study of diversity and classification of microbes, including those that cannot be cultured using traditional methods. Therefore, based on this sequencing technology, the microbial composition characteristics and significant biomarkers of perianal abscess can be comprehensively studied (Yin et al., 2023).

This study utilized high-throughput 16S rRNA gene sequencing to analyze bacterial communities in perianal abscesses and compared them with those in anal skin and fecal samples from the same individuals to understand the potential sources of infecting microbes. This study aimed to investigate the microbial sources in perianal abscesses, explore the origins and relationships of pathogenic bacteria, and provide guidance for prevent perianal abscess.

Materials and Methods

The flowchart of the experimental design used in this study was shown in Fig. 1. The studies involving humans were approved by Medical Ethics Committee of Qilu Hospital of Shandong University (Qingdao) under approval number KYLL-2023045. Written informed consent was obtained from all participants.

Figure 1 The flowchart of the experiment.

Study design and sample collection

Patients with a clear clinical diagnosis of perianal abscess, presenting with typical symptoms such as perianal pain, swelling, local redness, and possible fever, whose diagnosis is confirmed by a professional doctor through physical examination. Those who have not received any antibiotic treatment in recent 2 weeks or surgical drainage for the perianal abscess before the sample collection, to avoid the interference of previous treatments on the test results. All samples were collected from patients with perianal abscess who underwent surgery in the operating room of the proctology department at Qilu Hospital of Shandong University (Qingdao). With the participants’ informed consent, sterile cotton swabs were used to collect samples from the skin of perianal surface, feces, and abscess. For swab sampling of the abscess site, the skin should be disinfected before taking the pus sample. Some samples were obtained by puncturing the abscess cavity before the operation to get the pus, while others were collected by directly using cotton swabs to dip into the abscess cavity after the skin of the abscess cavity was incised during the operation. The collected samples were promptly placed in sample tubes containing 15% glycerin solution and stored at −80 °C until further analysis.

DNA extraction and 16S rRNA gene sequencing

Genomic DNA extraction was performed using the DNeasy PowerSoil Kit (Qiagen, Hilden, Germany), with modifications (glass beads were added into each sample tube and perform a short vortex. Then, centrifuge the tubes and discard the supernatant. After that, 750 microliters of buffer SA were added from the kit. Next, the instructions of the kit were followed completely for the subsequent operations). The quantity and quality of extracted DNA were assessed using several methods. A NanoDrop One spectrophotometer (NanoDrop Technologies, Wilmington, DE, USA) was used to measure the concentration and purity of DNA. Qubit 3.0 Fluorometer (Life Technologies, Carlsbad, CA, USA) was used to determine the precise DNA concentration. Agarose gel electrophoresis was performed to evaluate the integrity and size distribution of extracted DNA fragments.

Second-generation high-throughput Illumina sequencing was performed at Wuhan Benagen Technology Co., Ltd. The following is the experimental process of the sequencing company. PCR amplification of the V3–V4 region of the bacterial 16S rRNA gene was performed using the forward primer 341F (5′-CCTACGGGNGGCWGCAG-3′) and the reverse primer 805R (5′-GACTACHVGGGTATCTAATCC-3′). The thermal cycling conditions were as follows: an initial 98 °C for 45 s for DNA template denaturation. Then, 25 cycles of amplification, each with 98 °C for 20 s (denaturation), 60 °C for 30 s (annealing), and 72 °C for 50 s (extension). After that, a final 72 °C for 10 min for full fragment elongation. Post-PCR, agarose gel electrophoresis detected the amplified bands. PCR amplicons were purified using Vazyme VAHTSTM DNA Clean Beads (Vazyme, Nanjing, China). The library building kit was obtained from Vazyme: VAHTS® Universal DNA Library Prep Kit for Illumina. 16S rRNA gene sequencing was performed using the Illumina Novaseq 6000 platform in the PE250 mode (Illumina, Inc., San Diego, CA, USA).

Sequencing data analysis

After sequencing, the reads were de-multiplexed into samples based on barcodes and the sequences were imported into QIIME2 (v2021.4.0) (Bokulich et al., 2018). Raw data were filtered to remove adapter contamination and low-quality reads to obtain clean reads. Using the QIIME2 plugin vsearch (with default sequence similarity of 97%), operational taxonomic unit (OTU) classification was determined at the phylum, genus, and species levels. Reference alignment and taxonomic assignments were based on the SILVA database (v138.1) (Rognes et al., 2016; Quast et al., 2012).

Data analysis and statistics

The weighted UniFrac distance was calculated using the phyloseq package in R (v4.2.1). The Bray-Curtis distance was calculated using the vegan package in R. Hierarchal clustering was performed using the UPGMA. MaAsLin2 (Microbiome Multivariable Association with Linear Models v2.0) was used to identify biomarkers of the bacterial community using the linear models. ANOSIM and NMDS were performed using the vegan package in R. Two-sided Fisher’s exact tests were performed using the Python SciPy package. All other statistical analyses were performed using Jamovi 2.3.21.0 or Prism 9.4.1(681). Partitioning around medoids (PAM) was performed with weighted UniFrac distances using the cluster package on the R platform.

Results

Sample collection and analysis

To investigate the bacterial composition signatures of the abscess and to analyze putative origins of microbes leading to perianal abscess, samples were collected from the anal skin, feces, and abscess of 75 patients suffering from perianal abscess in 2023 (Fig. 2). The patients had an average age of 38.04, and a male-to-female ratio of 7.4:1. Samples were subjected to second-generation high-throughput Illumina sequencing, and bacterial community compositions were obtained for each sample. Patient information is shown in Table S1, in which samples ending with ‘−1’ indicates data from skin, samples ending with ‘−2’ indicates data from feces, and samples ending with ‘−3’ indicates data from abscess.

Figure 2 Sample locations.

Bacterial community compositions

Bacterial community compositions from the skin, feces, and abscess were determined and compared (Fig. 3), showing that, at the phylum level, Bacillota, Bacteroidetes, Proteobacteria, Actinobacteria, and Fusobacteroidota were the five most abundant phyla to which most bacteria belonged. The bacterial community in the skin contained substantially fewer bacteria from Bacteroidetes, whereas the abscess bacterial community contained substantially fewer bacteria from Bacillota. There is a comparable presence of Bacillota and Bacteroidota in the fecal samples. On the genus level, in skin samples, Bacteroides (11.95%), Prevotella (8.57%), Staphylococcus (7.75%), Escherichia-Shigella (5.80%), Corynebacterium (4.63%), Anaerococcus (4.57%), and Finegoldia (4.13%) are the most abundant genera; in feces samples, Bacteroides (21.67%), Escherichia-Shigella (15.82%), Prevotella (9.25%), Faecalibacterium (5.03%), and Megamonas (5.00%) are the most abundant genera; in abscess samples, Bacteroides (24.01%), Escherichia-Shigella (12.16%), Prevotella (8.63%), an Enterobacteriaceae genus (8.63%), Fusobacterium (8.12%), and Staphylococcus (3.45%) are the most abundant genera. Initial screening led to the observation of microbes of clear fecal origin, including Bacteroides and Escherichia-Shigella, as well as microbes of clear skin origin, including Staphylococcus in abscess samples.

Figure 3 Bacterial community compositions from the skin, feces, and abscess.

(A) Relative abundance at the phylum level. (B) Relative abundance at the genus level.

A comparison of alpha diversities between skin, feces, and abscess samples led to interesting findings (Fig. 4). Two-tailed paired t-tests were used for abscess samples, significantly lower Shannon indices (p = 8.28 × 10−9 for skin, and p = 1.54 × 10−5 for feces samples) and Simpson indices (p = 2.65 × 10−9 for skin, p = 1.72 × 10−6 for feces) were found for abscess samples, and only slightly lower diversities were found for feces samples compared with skin samples (p = 0.020 for Shannon index, p = 0.082 for Simpson index). These results suggest similar complexities of microbiome compositions between the skin and feces, yet significantly less diversified microbiomes in abscesses. Indeed, very simple microbiomes were found in abscess samples. For instance, for subject N159, 99.18% of the infecting microbes in the abscess were Staphylococcus. A similar case was not observed in skin or fecal samples.

Figure 4 Comparison of α-indices between skin, feces, and abscess samples.

Each dot represents a sample. Lines connect samples from the same subject. **p < 0.01; ***p < 0.001; ****p < 0.0001.

Indices indicating microbiome richness vary. No significant differences in both the Chao1 index (p = 0.17, two-tailed paired t-test) and ACE estimator (p = 0.07, two-tailed paired t-test) were found between the skin and abscess. However, significantly lower richness were found for feces than skin (Chao1 p = 1.68 × 10−3, ACE p = 2.24 × 10−3, two-tailed paired t-test) and abscess (Chao1 p = 1.75 × 10−3, ACE p = 4.15 × 10−4). This suggests a lower number of taxa from fecal samples, most probably because anal skin and abscesses receive microbes from both the skin surface and intestinal microbiomes, which may not change the complexity of microbiomes but increase the number of taxa.

The microbiomes of the abscess were subjected to a more detailed analysis with unsupervised clustering using the PAM method and microbial community profiles at the OTU level. Bray-Curtis distances were used for clustering. At k = 10, we observed 10 clusters with clear signatures (Table S2, Fig. 5). Clusters 1, 2, 6, 10 are Bacteroides-dominant types, with each cluster/subtype with distinct features: Cluster 10 is predominantly Bacteroides (average abundance 92.83%), Cluster 6 is primarily Bacteroides with also Escherichia-Shigella, Cluster 1 and 2 are both Bacteroides-Prevotella subtypes, while Cluster 1 is more complex. The other abscess microbiome types/clusters are each dominant with one or two specific genera: Cluster 3 is a Prevotella-Porphyromonas type, Cluster 4 is a Staphylococcus type, Cluster 5 is a Enterobacteriaceae type, Cluster 7 is a Fusobacterium type, Cluster 8 is an Aeromonas type, and Cluster 9 is an Escherichia-Shigella type. Cluster 4 had an average Staphylococcus abundance of 33.90%. Considering that Staphylococcus is a skin microbe, this cluster may originate from the skin. One-way ANOVA did not reveal an age difference in these abscess microbiome types (p = 0.056). Although the percentage of female patients (33.33%) in Cluster 3 appeared to be higher than the overall female percentage (12.00%), with a two-sided Fisher’s exact test, we were unable to find a significant enrichment (p = 0.070). Therefore, we were unable to find a significant correlation between patient information and abscess microbiome types.

Figure 5 Cluster analysis of microbiomes of the abscess with Bray-Curtis distances.

NMDS analysis with weighted UniFrac distances calculated between each microbiome revealed that microbiomes from anal skin, feces, and abscesses were significantly different (Fig. 6). The skin and fecal microbial communities are substantially distinct from each other, which was expected but needs to be tested here because anal skin is frequently contaminated with feces.

Figure 6 NMDS analysis of the microbiomes.

Further in-depth comparisons of skin and fecal microbial communities revealed biomarkers for each type of microbial community at the genus level (Table 1). Bacterial genera with at least 1% average abundance in either microbial community type had a prevalence (presence in percentage of samples) of at least 50%, and a fold difference of at least 1.5 were chosen for further analysis. The skin microbial community was enriched with Staphylococcus, Corynebacterium, Anaerococcus, and Finegoldia, along with other genera, whereas the fecal microbial community was enriched with genera including Bacteroides, Escherichia-Shigella, and Parabacteroides (Table 1). This is generally consistent with previously reported skin and fecal microbiomes (Cundell, 2018; Zafar & Saier, 2021; Li et al., 2022). It has also been reported that Anaerococcus and Finegoldia predominates buttock skins (Zheng et al., 2019).

Table 1 Comparison between skin and fecal microbial community.

Genera	Skin abundance	Feces abundance	Fold difference	FDR	
Biomarkers for skin microbial community	
Corynebacterium	0.046	3.14 × 10−3	14.77	1.83 × 10−23	
Staphylococcus	0.078	8.28 × 10−3	9.37	1.51 × 10−20	
Anaerococcus	0.046	7.22 × 10−3	6.33	2.30 × 10−12	
Finegoldia	0.041	7.14 × 10−3	5.79	3.22 × 10−10	
Peptoniphilus	0.028	4.70 × 10−3	5.92	8.52 × 10−9	
Acinetobacter	0.027	2.29 × 10−3	11.68	3.06 × 10−7	
Porphyromonas	0.027	2.87 × 10−3	9.51	1.87 × 10−4	
Pseudomonas	0.021	5.41 × 10−3	3.91	3.34 × 10−3	
Dialister	0.011	6.48 × 10−3	1.71	3.61 × 10−3	
Biomarkers for fecal microbial community	
Parabacteroides	8.77 × 10−3	0.027	3.10	3.53 × 10−4	
Escherichia-Shigella	0.058	0.158	2.73	5.10 × 10−4	
Bacteroides	0.120	0.217	1.81	5.73 × 10−3	
Lachnospiraceae genus	8.63 × 10−3	0.017	1.96	0.011	
Phascolarctobacterium	6.59 × 10−3	0.016	2.47	0.021	
Parasutterella	3.88 × 10−3	0.012	3.16	0.032	

A comparison between microbial communities from abscesses and the other two types of microbial communities led to several interesting findings. Skin microbial communities were enriched with highly abundant genera that are also biomarkers when compared with fecal samples, such as Corynebacterium, Staphylococcus, and Finegoldia (Table 2), showing significant bacterial composition signatures for skin samples. Meanwhile, we found that Staphylococcus, a skin biomarker, was enriched in abscesses compared to feces (Table 3). This clearly suggests that the abscess bacterial community is affected by the skin, prompting us to wonder whether skin microbes may be a source of abscess infection.

Table 2 Comparison between skin and abscess bacterial community.

Genera	Skin abundance	Abscess abundance	Fold difference	FDR	
Biomarkers for skin bacterial community	
Corynebacterium	0.046	5.45 × 10−3	8.50	5.40 × 10−12	
Staphylococcus	0.078	0.035	2.24	2.42 × 10−11	
Finegoldia	0.041	8.23 × 10−3	5.02	1.02 × 10−9	
Anaerococcus	0.046	0.012	3.83	1.96 × 10−8	
Peptoniphilus	0.028	0.014	1.95	1.44 × 10−5	
Enterococcus	0.022	3.63 × 10−3	6.19	5.05 × 10−5	
Blautia	0.012	6.36 × 10−3	1.95	1.14 × 10−4	
Subdoligranulum	0.014	4.16 × 10−3	3.47	1.46 × 10−4	
Bifidobacterium	0.012	2.81 × 10−3	4.29	5.42 × 10−4	
Collinsella	0.010	4.67 × 10−3	2.19	1.85 × 10−3	
Biomarkers for abscess microbial community	
Streptococcus	0.013	0.022	1.62	2.96 × 10−3	

Table 3 Comparison between feces and abscess bacterial community.

Genera	Skin abundance	Abscess abundance	Fold difference	FDR	
Biomarkers for fecal bacterial community	
Parabacteroides	0.027	4.68 × 10−3	5.81	1.79 × 10−11	
Lachnospiraceae genus	0.017	5.23 × 10−3	3.24	6.44 × 10−8	
Parasutterella	0.012	2.07 × 10−3	5.93	5.06 × 10−7	
Subdoligranulum	0.021	4.16 × 10−3	4.95	6.29 × 10−7	
Phascolarctobacterium	0.016	3.25 × 10−3	5.00	6.90 × 10−6	
Faecalibacterium	0.050	0.019	2.71	1.15 × 10−5	
Enterococcus	0.023	3.63 × 10−3	6.24	7.87 × 10−3	
Biomarkers for abscess bacterial community	
Staphylococcus	8.28 × 10−3	0.035	4.17	0.020	
Fusobacterium	0.012	0.056	4.57	0.034	

Analysis of putative sources of infecting microbes in abscess samples

To analyze putative sources of microbes in abscess samples, the three bacterial communities from the skin, feces, and abscess samples of each subject were analyzed and compared (Table S3). We found that in 26 samples, abscess microbes were likely from feces; in eight samples, abscess microbes were likely from the skin; 23 samples were likely from both the skin and feces; and in 18 samples, microbes were likely from neither source (Fig. 7). This suggests that multiple origins are present in infections that lead to perianal abscesses.

Figure 7 Proposed origins of abscess microbes.

Furthermore, a more in-depth analysis supports this hypothesis. Fifteen abscess samples could be considered to be dominated by one genus (one genus represents over 80% of all microbes). Cases in which this genus originated from the skin, feces, or other sources can be found within these 15 samples.

In three subjects, the microbes in the abscess clearly originated from the skin. In subject N112, 98.4% of the abscess microbes were Staphylococcus, a skin microbe comprising 51.8% of the skin microbial community. In subject N120, 95.9% of the abscess microbes were Enterobacteriaceae, comprising 87.8% of the skin microbial community. In subject N159, 99.2% of the abscess microbes were Staphylococcus, comprising 24.1% of the bacterial community in the skin.

In four subjects, the microbes in the abscess clearly originated from feces. In subject N140, 96.5% of the abscess microbes were Bacteroides which comprised 24.4% of the fecal microbial community. In subject N50, 95.2% of the abscess microbes are Escherichia-Shigella that comprises 21.7% of the fecal microbial community. In subject N54, 91.0% of the abscess microbes are Escherichia-Shigella that comprises 35.6% of the fecal microbial community. In subject N73, 87.6% of the abscess microbes were Bacteroides which comprised 30.5% of the fecal bacterial community.

In five subjects, it was difficult to determine the source of the abscess bacteria. In subject N111, 96.7% of the abscess microbes were Enterobacteriaceae, comprising only 0.16% and 1.20% of the skin and fecal microbial communities, respectively. In subject N113, 93.2% of the abscess microbes were Enterobacteriaceae, which comprised only 4.12% and 0% of the skin and fecal microbial communities, respectively. In subject N38, 88.6% of the microbes were Enterobacteriaceae, comprising only 0.20% and 0.90% of the skin and fecal microbial communities, respectively. In subject N69, 92.8% of the microbes were Bacteroides, which comprised only 0.40% and 0.93% of the skin and fecal microbial communities, respectively. In subject N96, 81.7% of abscess microbes were Enterobacteriaceae, comprising only 4.50% and 8.31% of the skin and fecal bacterial communities, respectively.

In many other subjects, abscess microbes have a clear skin-feces mixed origin. Some of the most significant cases include the following: in subject N105, Bacteroides which comprises 25.2% of abscess microbes, are from the feces (30.9% of fecal bacterial community), whereas Corynebacterium which comprises 11.1% abscess microbes, are from the skin (15.0% of the skin bacterial community). In subject N110, Staphylococcus which comprises 18.3% of abscess microbes, is from the skin (10.5% of the skin bacterial community), whereas Bacteroides which comprises 15.1% of abscess microbes, is from the feces (31.2% of fecal bacterial community). In subject N127, Prevotella which comprises 23.3% of abscess microbes, is from the skin (19.5% of the skin bacterial community), whereas Staphylococcus which comprises 21.1% of abscess microbes, is from the feces (36.2% of the fecal bacterial community). In subject N40, Bacteroides which comprises 14.3% of abscess microbes, were from the feces (40.9% of fecal bacterial community), whereas Porphyromonas which comprises 9.13% of abscess microbes, were from the skin (23.9% of the skin bacterial community).

These predictions were validated at the OTU level. By comparing the microbial community at the OTU level rather than at the genus level for samples in which a clear abscess microbe origin was predicted, we found, in general, the same prediction at the OTU level as at the genus level (Table S4).

Discussion

The above cases clearly verify the hypothesis that the infecting microbes in perianal abscesses may originate from the skin, feces, skin, feces, or other locations. These other locations may include the vagina, respiratory tract, and mouth. Given that Klebsiella pneumoniae, a member of the Enterobacteriaceae family, is indeed a fecal coliform commonly found in feces and can also be associated with ventilation—related respiratory tract infections, it is clear that the origin of the infecting microbes is more complex.

Another consideration that needs to be taken into account is that while this work used a quantitative microbial community for the analysis of microbial origins, it suffers from a possible drawback. Microbes can not only transfer from one location to another, but can also grow at different rates in different locations. Therefore, merely comparing the relative abundance of microbes may lead to mistakes. For instance, a low abundance of microbes in feces may grow faster on the skin. While its transfer from the feces to perianal abscess may be the de facto origin of infection, its abundance in feces may be lower than in the skin, therefore confusing the prediction of the origin of microbes in the abscess. In addition, the high abundance of a small number of microbes in the abscess may also be the result of bacterial growth in the abscess rather than a simple transfer from another location.

A previous culturing-based analysis of the origin of microbes in perianal abscesses suggested that infecting microbes primarily originate from the gastrointestinal tract (Chang et al., 2017). In this study, using a more comprehensive culturing-independent sequencing-based approach, we were able to identify the bacterial composition signatures of perianal abscesses and analyze the origins of abscess microbes at a higher resolution. Interestingly, while the gastrointestinal tract is still a primary source of abscess microbes, other niches in the human body can also be sources of abscess microbes. The transmission of microbes from these niches may involve more complex pathways and behaviors than previously acknowledged, and further studies are needed to elucidate the etiology of perianal abscesses.

Conclusions

The skin, fecal, and abscess bacterial communities were studied in 75 patients with perianal abscess to determine the bacterial composition signature and origin of infecting microbes in the abscess. The three types of bacterial communities exhibited significant differences in composition, with the abscess bacterial community displaying lower diversity and the fecal bacterial community exhibiting lower richness. Further comparison revealed significant biomarkers from each type of bacterial community, such as Staphylococcus and Coryne-bacterium in skin samples, and Bacteroides and Escherichia-Shisgella in fecal samples. With the analysis of the abscess bacterial community, it was clear that abscess microbes may originate from the skin, feces, skin, feces, and other locations. This finding reveals the complexity of the origin of microbes in perianal abscesses, which benefits further etiological studies of this disease. In this study, we aimed to analyze the genetic variation of isolates at various loci. The source of the bacteria was further analyzed based on genetic information, plasmids, and other factors.

Supplemental Information

Supplemental Information 1 Patient information.

Supplemental Information 2 Clustering of abscess microbiomes.

Supplemental Information 3 Tracking prominent source of microbiomes in anal abscesses.

Supplemental Information 4 Prediction of microbe source in abscess on OTU level.

Additional Information and Declarations

Competing Interests

Author Contributions

Human Ethics

Data Availability

The authors declare that they have no competing interests.

Song Han conceived and designed the experiments, prepared figures and/or tables, and approved the final draft.

Wenya Su conceived and designed the experiments, prepared figures and/or tables, and approved the final draft.

Kefeng Fan performed the experiments, authored or reviewed drafts of the article, and approved the final draft.

Zhen Xu performed the experiments, authored or reviewed drafts of the article, and approved the final draft.

Hai Xu analyzed the data, authored or reviewed drafts of the article, and approved the final draft.

Mingyu Wang analyzed the data, authored or reviewed drafts of the article, and approved the final draft.

Ling Li conceived and designed the experiments, prepared figures and/or tables, and approved the final draft.

Wenlong Shen analyzed the data, authored or reviewed drafts of the article, and approved the final draft.

The following information was supplied relating to ethical approvals (i.e., approving body and any reference numbers):

Medical Ethics Committee of Qilu Hospital of Shandong University (Qingdao) under the approval number KYLL-2023045.

The following information was supplied regarding data availability:

The data is available at Genome Sequence Archive: CRA013057, CRA014090.

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
