# Peer review of "The bacterial composition signatures of perianal abscess and origin of infecting microbes"

_PeerJ, doi:10.7717/peerj.18855_

## Round 0.1 · original submission · Major Revisions

I would propose to the authors trying to convince reviewer 2 that the findings are relevant from the biological point of view. The authors should try to discuss "the real problem of the article is the absence of the distinction between perianal abcesses and anal fistula". Reviewer 1 main comments are related to absence of a proper statistical work. It MUST be done!

Reviewer 1 ·

Basic reporting

This study explored the microbial types causing perianal abscesses. The author revealed microbes from feces and anal skin are sources of infection. I have some comments for the authors to improve their manuscript.
Introduction part
1. Line 42-47: The author need to explain (shortly) for the mechanism of anal abscesses.
2. Line 54: Please describe an examples of non-physical risk factors?

Experimental design

1. Inclusion and exclusion criteria should be mentioned.
2. Line 91: Brief description is needed for DNA extraction with some modifications.
3. I recommend the author to make a flowchart for the study design.
4. PCR amplification condition should be mentioned.
5. I think that the author may check the PCR products by gel electrophoresis before purification, so, it should be described.
6. A brief description of Illumina sequencing should be done. Except the author send to the company to sequencing. If so, you need to specify.

Validity of the findings

1. What are the underlying conditions of the patients?
2. Line 137-139: It lacks bacterial community in feces. This should be mentioned.
3. Line 139-148: Is there statistical significance among these bacterial types from 3 origins?
4. Line 167-183: Is there some statistical significance between these clusters and some patient's characteristics?
5. Are some patients with bad perianal conditions related to microbial types?
6. Line 263: K. pneumoniae is a one of the fecal coliforms, it can be found in feces. In the case of respiratory tract infection, it might occur from ventilation-associated. So, I do not agree from the sentence on lines 262-264.
7. Tables 1-3: Can the author show statistical significance from comparison?

Reviewer 2 ·

Basic reporting

The study aimed at looking at the bacterial composition of perianal abcesses.
The design of the study was done correctly but the real problem of this study is its aim.
Indeed, perianal abcesses are a common condition. They are frequently caused by anal fistulas but sometimes, there are abcesses related to sebaceous cycts or Verneuil's disease
The fistulas are caused in over 90% of cases by the infection of a cryptic gland in the pectinate line. In the other 10% of cases, it can be related to crohn's disease, tuberculosis,etc.

The real problem of the article is the absence of the distinction between perianal abcesses and anal fistula.

Moreover, detecting the bacterial composition of these abcesses would not very helpful because the treatment is always surgical or done via an incision during a consultation. Antibiotics are not always necessary.

Beside that, knowing the composition of the bacteria is not helpful in the clinical setting.

Experimental design

The study design is good but it was not mentionned how the specimen was taken from the abcess.
Indeed, if the incision was done throughout the skin leading to the culture, there will be a contamination of the skin. This can explain potential bias in the results of the culture and leads to confusion.

Validity of the findings

The findings are interesting to read but in the clinical setting, I don't think that they are relevant and will not help to understand better the pathophysiology of perianal abcesses.

---

## Round 0.2 · accepted · Accept

I am completely satisfied by the way that the authors took care of all comments and requests of the reviewers.

Reviewer 1 ·

Basic reporting

None

Experimental design

None

Validity of the findings

None

Additional comments

The author have response my comments completely. I have no further comments.